


# CycloneDetector (v1.0) - Algorithm for detecting cyclone and anticyclone centers from mean sea level pressure layer

Martin Prantl[1], Michal Žák[2], and David Prantl[3]

[1]Department of Computer Science and Engineering, Faculty of Applied Sciences, University of West Bohemia
[2]Department of Atmospheric Physics, Faculty of Mathematics and Physics, Charles University
[3]Ventusky project, InMeteo, s.r.o., Plzeň, Czech Republic

**Correspondence:** Martin Prantl (perry@kiv.zcu.cz)

**Abstract.** Automatic methods for identifying and tracking cyclones were firstly constructed in 1990's and since then there was a big increase in a precision and probability of detection. These methods have been traditionally focused on cyclones (and particularly on tropical cyclones), but the question of anticyclone centers detection remained unsolved since they are usually not a source of turbulent weather, precipitation etc. However, this issue can be important in the era of the climate

change. In this paper, an algorithm for an automatic detection of both, cyclones and anticyclones based on mean sea level pressure field, is presented. The algorithm uses two-dimensional raster data as an input and returns a list of detected pressure systems. The main advantages of our solution are easy implementation since it is based on the standard image processing algorithm, sufficient performance of the algorithm, and especially the possibility of high-pressure systems detection. Moreover, the presented solution does not need a direct terrain filtering needed for some algorithms to be done. To validate the quality

of detection algorithm results, a comparison against manually prepared data by *Met Office* was used. It follows from the comparison that the presented algorithm produces results similar to those by *Met Office*. The most significant differences can be found in the detection of cyclones at the beginning or the end of the lifespan stage. *Met Office* detects more cyclones in these stages than the presented solution.

## 1 Introduction

Pressure systems are the main features of atmospheric circulation. Cyclones and anticyclones (or lows and highs) affect daily weather over the whole Earth, and they are the main drivers of surface wind. Cyclones (anticyclones) are areas of relatively low (high) pressure compared to its surrounding, usually having enclosed isobars (on the surface pressure field) or izohypse (at the higher pressure level), they are, based on convention, divisible by 5 hPa, or 4 gpdam (geopotential decameters). Especially cyclones are essentially important for transporting heat and energy between tropical and polar regions. Therefore, it is useful

to have a profound knowledge of cyclones and anticyclones: their behavior, occurrence and movement. Especially in the era of deepening climate change, it is important to study changes in their frequency in different parts of the world.

For this reason, we need to have a robust algorithm for detecting and tracking pressure systems. While we can find many papers on automatic tropical cyclone positioning and detecting, papers on the detection of extra-tropical cyclones are rare, especially those elaborating on methods used in operational praxis. The reason is caused by an urgent need to find the tropical





cyclone center connected with very pronounced weather and its changes. Last but not least, the tropical cyclones can be detected more easily due to their regular structure with a quasi-circular closed center compared to extra-tropical cyclones (e.g., Wang et al., 2020), but as our paper is focused on extra-tropical systems, tropical cyclones will not be discussed in details.

Some decades ago, the detection of pressure systems had traditionally been based on a manual analysis of synoptic (paper) weather charts. Although this method is accurate, it is connected with a huge amount of time-consuming manual work. It has

to be pointed out that in data-scarce areas (e.g., over oceans or desert regions), the extrapolation does not necessarily lead to accurate detection of pressure systems centers. This time-consuming method could also have another source of problems, e.g., inconsistencies that may have arisen in the identification of cyclone and anticyclones centers by various analysts from day to day. Later, with the onset of the enhanced numerical models and better observational techniques, this lack-of-data disadvantage has been minimized, but the need for automated detection of lows and highs centers has appeared. Automated schemes have

started to be used since the 1990s and these schemes have overcome the above described drawbacks of manual analyses.

One of the first automatic schemes was introduced by Murray and Simmonds (1991) with a focus on mid-latitude cyclones in the Southern Hemisphere. Other automatic methods for identifying and tracking cyclones were constructed during the end of the 20[th] and the beginning of the 21[st] century (e.g., Hodges, 1994; Blender and Schubert, 2000; Rudeva and Gulev, 2007; Raible et al., 2008; Hanley and Caballero, 2012; Lu, 2017). These methods could benefit from new observation technologies

as well as from the development of numerical weather models.

To detect cyclones, various meteorological parameters and criteria were used in the past, typically, pressure or vorticity parameters. Searching minima in the sea level pressure and/or in the geopotential height at pressure levels can be found in more studies (e.g., Geng and Sugi, 2001; Rudeva and Gulev, 2007; Hanley and Caballero, 2012). In addition, some papers focus on finding minima of a geopotential height of the 850 hPa level (Kew et al., 2010), using relative vorticity of 850 hPa

at pressure levels (Flaounas et al., 2014), or a combination of these parameters (e.g., mean sea-level pressure minima and low-level vorticity maxima (Simmonds et al., 2008)). A very comprehensive overview intercomparing extra-tropical cyclone detection and tracking algorithms before 2013 can be found in the work of Neu et al. (2013). This overview poses many interesting questions and highlights problems that may have arisen when constructing the detecting algorithms, e.g., detecting cyclones over elevated topography that has not been standardized. Various methods deal with this point differently. Another

approach for detecting the cyclone centers is represented by a deep learning approach, first presented by Hinton et al. (2006). This approach can be based either on classical convolutional neural networks alone or in combination with a region proposal network. Recently, Lu et al. (2020) presented a mask region-based convolutional neural network for detecting the cyclones showing higher efficiency, especially for recognizing shallow or moderately deep cyclones of subsynoptic scale.

Generally, the scheme of cyclone center detection comprises several steps. Firstly, the center of the anti/cyclone has to be

detected. This can be done by several methods, such as finding the local pressure maximum or minimum, or searching the Laplacian of the pressure or relative vorticity. The detection of the anti/cyclonic center can be done by various methods. For example, Lu (2017) searched for a point lower than surrounding grid points (in all directions) on a latitude-longitude mesh in the grid of a mean sea-level pressure or a geopotential height of the 850 hPa level. Jiang et al. (2020) introduced the eight-section slope detecting method, based on mean sea-level pressure or geopotential height. Usually, several iterations are required





over an enclosed contour search stage to conduct the procedure for every cyclone-center candidate. After the center is detected, its pressure value and other parameters like the cyclone's depth, size, or radius can be obtained.

Finally, it is important to know what the cyclone centers detection accuracy really is. Generally speaking, no identification algorithm can guarantee 100% accuracy (Jiang et al., 2020). A large fraction of errors in detecting cyclone centers is caused by mistaking a trough for a closed cyclone. Other discrepancies between manually and automatically detected cyclones can be

connected with the value taken as the interval for detecting enclosed contours. It can be expected that the number of identified lows increases as the interval decreases (Wernli and Schwierz, 2006). Especially small and/or shallow cyclones are mostly sensitive to the choice of the interval. The denser the counters become, the more relatively small-scale weather systems will appear, while the original systems remain the same, despite the potential extensions of their original boundaries (Lu, 2017). It is not easy to find the accuracy of detection methods. While some authors (e.g., Jiang et al., 2020) provide approximately 85%

accuracy compared to manual analysis, these numbers are usually not confirmed by other studies. Qualitative conclusions are given instead. As Neu et al. (2013) states, the largest differences when comparing various methods and schemes can be found in the frequency distributions for short-lived, shallow, and slowly moving cyclones, especially on the northern hemisphere than on the southern hemisphere. This method's sensitivity should be beard in mind when using a single method. This is especially true for the total cyclone counts and the number and role of weak cyclones in the statistics.

When dealing with mid- and high-latitude pressure systems, sometimes the very miscellaneous structure of extra-tropical cyclones and especially anticyclones need to be processed by algorithms for their automatic detection. Relatively regularly shaped pressure patterns are typical for very deep lows or intense highs, characteristically occurring in the initial development stages. However, complicated structures with more centers can be detected during the mature and dissipation stages. Algorithms for automatic detection have to deal with various sizes (hundreds to thousands of kilometers), depths (from 920 to 1080 hPa),

or translational velocities (0 to 100 km/h) of the pressure systems. The general problem to be solved by automatic algorithms is the accurate detection of an outer anti/cyclonic boundary because it is not easy to find a set of closed counters. The detection of short-lived systems (such as thermal lows) can also be quite challenging for some widely used algorithms. Some schemes use a minimum life cycle time in order to omit such systems (see e.g., Hanley and Caballero, 2012). Nonetheless, such an approach can cause the omission of important weather features (e.g., heavy precipitation). For instance, this can be brought

by short-lived cut-off cyclones. Another problem can arise with open systems, i.e., local pressure maxima/minima without any closed isobars/isohypse. These include lee cyclones or luv anticyclones. It can also happen that algorithms were designed to search for cyclones of specific types (e.g., polar lows, extremely deep cyclones). However, these are not able to detect all other cyclones virtually in various latitudes.

The main reason for automatic detection of pressure patterns is to have an instrument for a great number of (re)analyses:

detecting lows and highs manually would be too time-consuming. However, to have this automatic detection available, climatology of various anti/cyclone characteristics can be prepared. This makes it possible to analyze parameters such as spatial distribution, evolution characteristics or frequency of occurrences in various regions (e.g., Dacre and Gray, 2009; Lu, 2017). This is especially valuable when studying long-term data with respect to the climate change.





In this paper, we focus on the surface pressure level field. As will be explained and discussed in details further, the main
features of our proposed solution are:

- – automatic detection of low- and high-pressure systems

- – low number and predictable behavior of input parameters

- – fast and easy to implement solution

In Chapter 2 we provide a detailed explanation of our solution. Chapter 3 presents the algorithm results. Our conclusion and
final discussion can be found in Chapter 4.

## 2   Proposed algorithm

The proposed algorithm detects low- and high-pressure systems directly from the mean sea level pressure layer (MSLP). This
quantity is widely available as an output from numerical weather prediction models (NWP), such as ECMWF, GFS, ICON,
and HRRR. The algorithm uses two-dimensional raster data as an input. It returns a list of detected pressure systems. Generally
speaking, there is a need for 2D Earth projection to be used for the raster data. The projection provided directly by the generated
NWP is used in our solution because the algorithm does not depend on any particular projection directly. Typically, inputs in
the equirectangular projection are used. Nevertheless, the Lambert-Conic projection for data from regional models such as
HRRR can also be applied.

The proposed solution consists of two steps: the detection of candidates (see Subsection 2.1) and the pressure system center
detection within these candidates (see Subsection 2.2).

### 2.1   Candidates detection

In the first step of the algorithm, the isobars with a given step, $C$, between pressures are found. The size of the step $C$ can be set
by the user. Most of the low-pressure systems are in an intensity range $1 - 15\ hPa$ (see Rudeva, 2008). Therefore, the isobars
are usually drawn with steps of 1, 2, 4, 5, or $10\ hPa$. We use the same steps for low- and high-pressure systems. With lower
values, we can detect more systems. Our recommendation for global models is to use 2 or $4\ hPa$. For local models, capturing
a smaller area, we recommend to use $1\ hPa$. From the field of isobars, we select a list of candidates, i.e., the potential centers
of pressure systems.

Detecting isobars from raster data of size $W \times H$ is a straightforward process. We use the simple Marching Squares (Lorensen
and Cline, 1987) algorithm. It is a 2D version of the well-known Marching Cubes algorithm for iso-surface extraction. The
detected isobars form enclosed polylines and have a fine, pixel-based, resolution. For further computations, these polylines can
be simplified (e.g., using Douglas and Peucker, 1973) to improve the performance of the following steps.

From the generated isobars, we select the pressure system candidates. They do not have any other closed isobars. An example
can be seen in Fig. 1. There are two main candidates (#1 and #2) and several small areas that are clearly not supposed to be
indicated as a pressure system centers. Therefore, these small areas need to be removed.

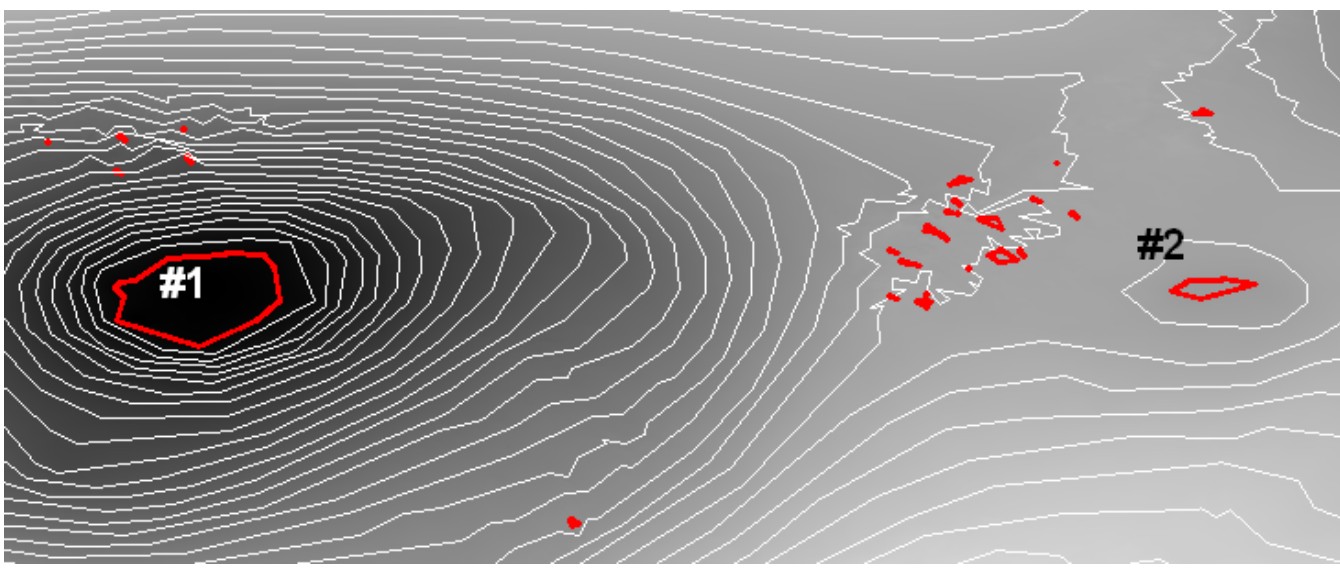

**Figure 1.** Candidates for isobars (red thick lines). There are no other closed isobars inside them. #1 and #2 are expected candidates for a pressure system. Other candidates are incorrect. Data are from a segment over Europe taken from ICON-EU model, 2021-10-24 12:00 UTC.

To remove unwanted areas, we use filtering based on the area of the contour area. For each candidate, we calculate its axis-aligned bounding box (AABB) that is used as a rough approximation of the contour area. However, the area size cannot be easily represented in image pixels because of the projection distortions that can be visualized with Tissot's indicatrix (Goldberg and Gott, 2007). Based on this observation, we use the area size in $km^2$. For this, the projection of the input image must be known and the inverse projection formulas are used to convert pixels positions $[x,y]$ to GPS coordinates $[lon, lat]$. From the

GPS coordinates of AABB corners, the area is approximated with equations based on Chamberlain and Duquette (2007).

     For filtering, the area size threshold, $T$, is used. The threshold value can be adjusted by the user. The value depends on the average area that the pressure system occupies. We recommend to use the values between $10,000 km^2$ and $20,000 km^2$.

     If the area size is smaller than the selected threshold, we have two possibilities:

   – remove the area.

– change the candidate to the nearest larger detected contour (called the "parent").

The second option is done if the "parent" area is smaller than the $M$ multiple of the threshold $T$ (based on our experiments, we selected $M = 100$).

     The process is repeated until there is a change in the candidates. The result of this filtration step, applied to Fig. 1, can be seen in Fig. 2. The small areas are entirely removed. Candidate #2 has changed from the inner contour to its "parent". After

this change, small areas with significant pressure changes are preserved that would otherwise be removed.



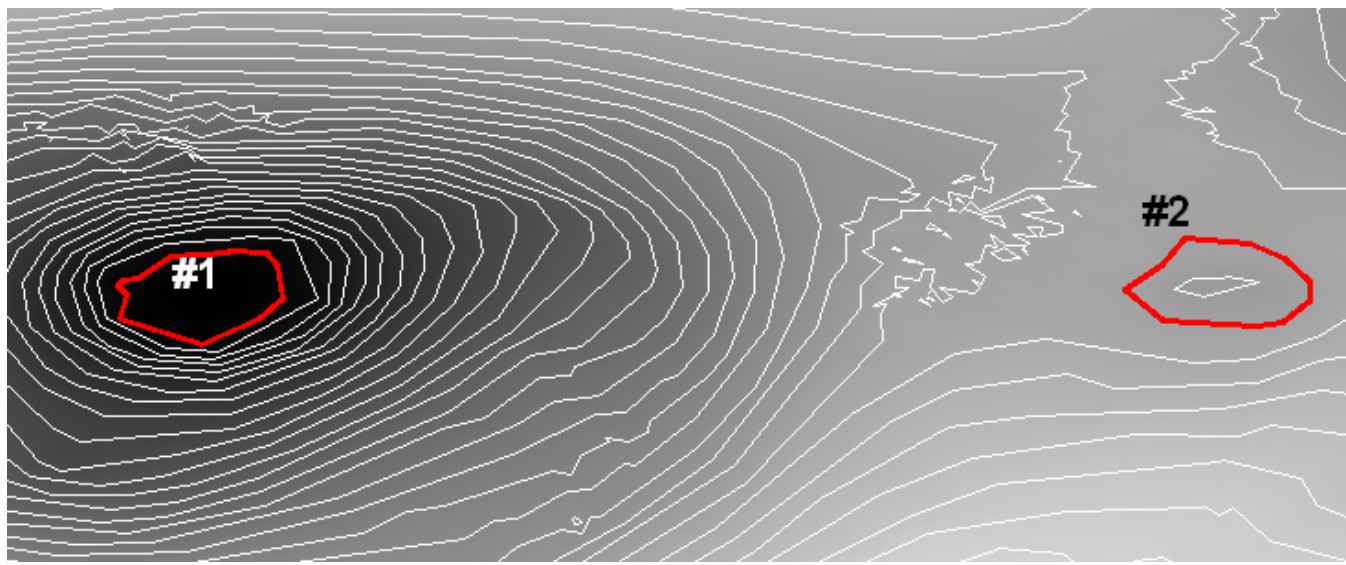

**Figure 2.** Candidates for isobars (red thick lines) after filtration. Small areas were either removed, or the candidate was moved to the "parent" contour (compare candidate #2 in Fig. 1 and in Fig. 2). Data are from a segment over Europe taken from ICON-EU model, 2021-10-24 12:00 UTC.

## 2.2 Center detection

The detection of pressure system centers is based on finding local extrema in input raster data. Only discrete data are available for our purposes. A possible solution would be to create an interpolant and search for extrema directly from the interpolated function analytically. However, this is a slow and complex task. Instead, we apply convolution-based operations directly on discrete data.

Using the Sobel operator (see Kanopoulos et al., 1988), we calculate the first derivative $dx$ and $dy$ of the input raster data. To identify the extrema, we only need to find the signs of the derivatives ($signDx$ calculated from $dx$ and $signDy$ calculated form $dy$).

### 2.2.1 Low-pressure system

To determine if a pixel represents a low-pressure system, we use its neighborhood and compare every pixel against the mask with the radius $r$ (therefore, the width and height are $2r+1$). For simplicity, we use square masks with patterns shown in Table 1. The size of the radius $r$ is an input parameter that can be adjusted by the user. However, based on our experiments, the default size of the radius, $r$, can be auto-calculated. The size is the average square root of the pixel size from the pressure system candidates AABB's (see Subsection 2.1). For high-resolution inputs, this can lead to large radius values and consequently to longer computations. If this is an issue, the radius can be set manually. Usually, based on our experiments, the size of radius selected from range $(5, 50)$ pixels is sufficient based on the input resolution. For a higher resolution, a larger radius is recommended.





**Table 1.** Example of masks for local minimum detection. The left mask for $dx$, the right mask for $dy$. For this example, the mask radius $r$ is set to be 2.

$$
\begin{bmatrix}
-1 & -1 & 0 & +1 & +1 \\
-1 & -1 & 0 & +1 & +1 \\
-1 & -1 & 0 & +1 & +1 \\
-1 & -1 & 0 & +1 & +1 \\
-1 & -1 & 0 & +1 & +1
\end{bmatrix}
\qquad
\begin{bmatrix}
-1 & -1 & -1 & -1 & -1 \\
-1 & -1 & -1 & -1 & -1 \\
0 & 0 & 0 & 0 & 0 \\
+1 & +1 & +1 & +1 & +1 \\
+1 & +1 & +1 & +1 & +1
\end{bmatrix}
$$

For every pixel $[x, y]$ that is inside the candidate obtained in Subsection 2.1, we apply a mask to $signDx$ and $signDy$. If the signum value corresponds to the mask value, the pixel is marked as correct (the counter $okX$ and/or $okY$ is increased). Finally, the ratio of correctly marked pixels $okX$ and $okY$ against the area size of the candidate is calculated. If the ratio of correct pixels is above $60\%$, the pixel $[x, y]$ is marked as extreme. The process can be seen in Algorithm 1.

160

---

**Algorithm 1** Pseudo-code for testing whether the pixel (x,y) is an extrema candidate. Variable $r$ is the mask radius. The code does not handle image borders

---

1: $maskArea \leftarrow (2 * r + 1) * (2 * r + 1)$
2: $okX \leftarrow 0$
3: $okY \leftarrow 0$
4: **for** (yy = y - r; yy <= y + r; yy++) **do**
5:    **for** (xx = x - r; xx <= x + r; xx++) **do**
6:       $maskSign \leftarrow mask[yy - (y - r)][xx - (x - r)]$
7:       **if** $maskSign \leq signDx[yy][xx]$ **then**
8:          $okX \leftarrow okX + 1$
9:       **end if**
10:       **if** $maskSign \leq signDy[yy][xx]$ **then**
11:          $okY \leftarrow okY + 1$
12:       **end if**
13:    **end for**
14: **end for**
15: **if** $okX/maskArea \geq 0.6$ **and** $okY/maskArea \geq 0.6$ **then**
16:    pixel (x, y) is extrema
17: **end if**

---

Running the algorithm 1, extrema areas within candidate contours are obtained as can be seen in Fig. 3.

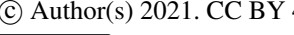


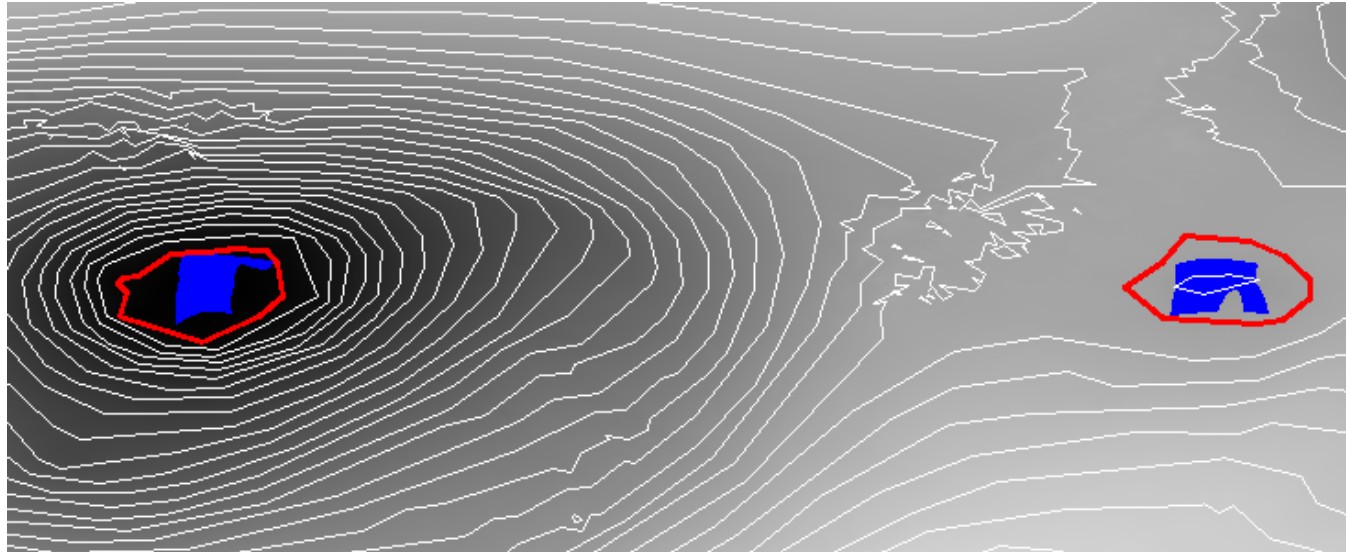

**Figure 3.** Located extrema pixels after applying the masks. Data are from a segment over Europe taken from ICON-EU model, 2021-10-24 12:00 UTC.

Sometimes, there can be multiple isolated areas detected inside a single candidate. This is caused by numerical problems. To overcome this, we select the largest area from all isolated areas inside a single candidate and find its centroid. This centroid

165    is used as the location of the pressure system.

### 2.2.2   High-pressure system

High-pressure systems are detected with the same algorithm as low-pressure systems in 2.2.1. The only change is in the masks. We need to find local maxima and. Therefore, we use a mask with swapped signs (cf., Table 2).

**Table 2.** Example masks for local maximum detection. The left mask for $dx$, the right mask for $dy$. The mask radius $r$ is set to be 2.

$$
\begin{bmatrix}
+1 & +1 & 0 & -1 & -1 \\
+1 & +1 & 0 & -1 & -1 \\
+1 & +1 & 0 & -1 & -1 \\
+1 & +1 & 0 & -1 & -1 \\
+1 & +1 & 0 & -1 & -1
\end{bmatrix}
\quad
\begin{bmatrix}
+1 & +1 & +1 & +1 & +1 \\
+1 & +1 & +1 & +1 & +1 \\
0 & 0 & 0 & 0 & 0 \\
-1 & -1 & -1 & -1 & -1 \\
-1 & -1 & -1 & -1 & -1
\end{bmatrix}
$$

### 2.3   Filtering results

170    In some cases, it may happen that low- and high-pressure systems are detected together inside a single candidate.





This is often caused by multiple shallow systems inside a single candidate. In this case the algorithm is set to select a larger system. We can also use weighting based on the distance of the centers of the systems from the center position of the candidate contour.

Centers that are too close to each other might cause another problem. If this is a problem, we may apply the optional, user-defined threshold distance $D$ in $km$. In this case, if two systems are closer than the threshold $D$, the system with a smaller area is removed.

## 2.4 Performance

The performance of the algorithm depends on the resolution of input raster data and on the number of detected pressure systems. The more systems have been detected, the slower the algorithm is.

Algorithm performance can be influenced by detection if a pixel is inside a candidate (see Subsection 2.2.1). In the naive solution, we have to test if a pixel is inside the polygon for every candidate. This is rather slow. However, if a bounding volume hierarchy created from candidates of AABBs is used (e.g., Pharr et al., 2016), we can improve the speed of the algorithm considerably. In this case, the point in AABB is tested first. Therefore, a computationally expensive test to determine whether a pixel is inside the polygon is performed only for several contours.

## 2.5 Limitations

The proposed solution has its limitations in the input projection. It follows from the projection of the input raster data, that not all the systems may be detected. For example, in the case of equirectangular projection, systems near the poles may be detected incorrectly. Hence, instead of one center, several or no centers may be obtained. This problem is caused by a distortion near the poles. Data reprojection might lead to a solution if one projection for low-distorted latitudes is used, which may be from the interval $(-85°, 85°)$. A different projection is only applied for polar regions.

There might be another minor limitation in the selection of the input parameters, mainly the area size threshold $T$. However, once the parameters have been set, they can be reused for the same NWP model in most cases.

## 3 Experiments and results

In this section, a comparison of the proposed solution with a manually created list of pressure systems is provided.

## 3.1 Algorithm setup

To evaluate the proposed solution, we have used historical data obtained from the reanalysis of the ECMWF model (dataset ERA5, Hersbach et al., 2021). We have run the proposed algorithm with the following settings as shown in Table 3.





**Table 3.** Experiment settings. Mask radius $r$ is auto-calculated from area threshold $T$. Minimal center distance is disabled.

| Parameter | Value |
|---|---|
| Isobars step $C$ | $2\ hPa$ |
| Area threshold $T$ | $10'000\ km^2$ |
| Mask radius $r$ | automatic |
| Minimal center distance $D$ | $0\ km$ |

The relation between input parameters, area threshold $T$ and isobars step $C$, is depicted in Fig. 4. The results are for a global model (ICON) from a single time, 2021-03-18 12:00 UTC. It can be observed that for a majority of parameters combinations

we obtain a similar count of pressure systems in the interval $(75, 125)$. On the other hand, for the small values of area threshold and isobars step we obtain too many pressure systems with a lot of false detections.





**Figure 4.** Relation between input parameters, area threshold $T$ and isobars step $C$, with respect to the number of detected pressure systems.

We recommend to use auto-calculated values of mask radius $r$ from Table 3. For comparison, we present some auto-calculated values in Table 4. The values are obtained from multiple runs over different times. Results are averaged and rounded to whole numbers to represent pixels.





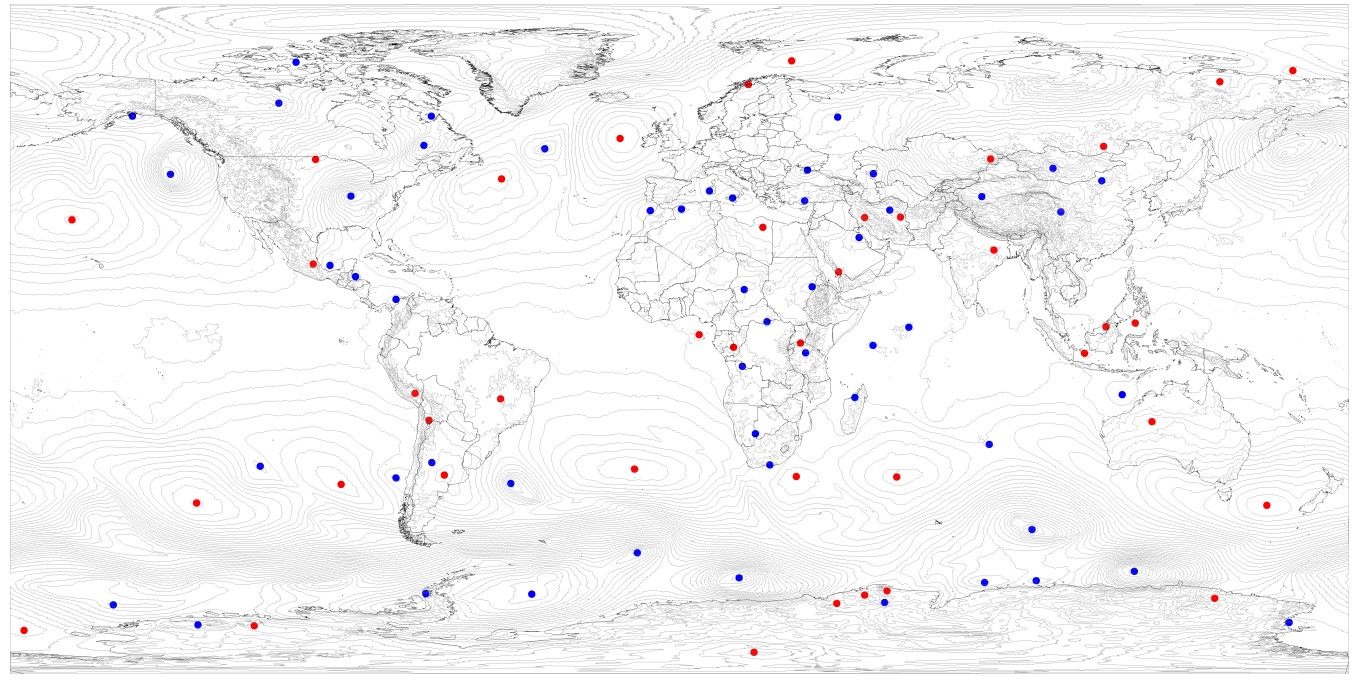

**Figure 5.** The detected pressure systems from ICON model based on Table 3. The red dots indicate high-pressure systems, and the blue dots are low-pressure systems. Data from 2021-03-18 12:00 UTC.

**Table 4.** Auto-calculated values of mask radius $r$ in pixels for some NWP models with a standard deviation $\sigma$ in [pixels].

| Model | Coverage | Resolution | $r$ [pixels] | $\sigma$ [pixels] |
|-------|----------|------------|--------------|-------------------|
| ECMWF | Global | $1440 \times 721$ | 9 | 1.4 |
| ICON | Global | $2880 \times 1441$ | 14 | 2.1 |
| ICON | Europe | $1094 \times 657$ | 28 | 4.3 |
| HRRR | USA | $1799 \times 1059$ | 55 | 7.7 |

An example of detected pressure systems from global ICON model with settings from Table 3 can be seen in Fig. 5.

### 3.2 Evaluation

The evaluation of the proposed solution is conducted on selected datasets from 1998 to 2020. We have used datasets from Europe and the North Atlantic ocean region provided by *Met Office* and available at MetOffice (2021); Müller and Floors (2021). There are pressure centers (highs and lows) available as being created and checked by the national meteorological

institute in the UK. The quality of widely used data is high. However, they are only available as images. Thus, we have to process the data manually and create a list of the centers of pressure systems.

Since it is considerably time-consuming to process all images manually from the available range, we have created only a random subset. Its size was determined based on Cochran (1977). For a $95\%$ confidence level, we have to determine a sample of 385 images. Based on this observation, to determine the precision of the proposed solution we have randomly selected 385

pressure lows and highs (with the same ratio) from 1998 to 2020. Our detected pressure system is the same one as in the *Met Office* data if both systems are of the same type and their centers are closer than 300km to each other (the limit is set higher because of the differences in the source and the origin of the datasets). Otherwise, our detected system is marked as different.

In the first selection, $n_1$, the pressure systems were determined by our solution and compared against data from *Met Office*. The results are shown in Table 5. It follows from the data that $92\%$ of the pressure systems detected by our algorithm were also

detected in the *Met Office* analysis.

**Table 5.** Comparison between the proposed solution and the *Met Office* analysis

| Sample size | Same | Different |
|:---:|:---:|:---:|
| 385 | 354 (92%) | 31 (8%) |

In the second selection, $n_2$, the pressure systems were selected from *Met Office* datasets and compared against the systems detected by our proposed solution. The results are shown in Table 6. It follows from the data that $83\%$ of pressure systems from *Met Office* are found by our solution.

**Table 6.** Comparison between the *Met Office* analysis and the proposed solution

| Sample size | Same | Different |
|:---:|:---:|:---:|
| 385 | 320 (83%) | 65 (17%) |

The results from Tables 5 and 6 together with Cochran (1977) provide the following conclusion: if a low- or high-pressure

system were detected by our algorithm, the same system would be found with a probability of $92\%$ by *Met Office*. Conversely, if the system center were detected in *Met Office* data, it would be found in our algorithm output with a probability of $83\%$. The difference is caused by the *Met Office* methodology. If we consider low- and high-pressure systems separately, we can observe similar probabilities of the results.

Three main reasons for differences between the systems detected from our solution and *Met Office* can be recognized.

First, the proposed solution is universal. Hence, it can be used for the entire world, whereas *Met Office* has traditionally focused on Europe and the North Atlantic region. This, for example, enables identifying cyclones at a very early stage.



The second reason is time continuity. The proposed solution aims at detecting low- and high-pressure systems from a single input. *Met Office* takes the previous and the following time steps into consideration (time consecution is conserved). For example, filling up low-pressure systems can be detected manually for a longer period of time by meteorologists from *Met*
*Office*.

The third reason is the data itself. We use reanalysis from the ECMWF that can be slightly different from the *Met Office* data.

### 3.2.1 Comparison with other methods

Comparison between the proposed solution and *Met Office* shows similarity $92\%$ as we stated above. However, it is also
important to demonstrate how other automated algorithms are accurate.

One of the latest algorithms for the detection of cyclones was published by Jiang et al. (2020). They use the mean sea level pressure (MSLP) or geopotential height to identify extra-tropical cyclones. The accuracy is measured on a sample of $500$ cyclones, which are determined by the algorithm and compared with manually determined data. The average match is around $85\%$. Thus, the results are similar to our proposed algorithm.

Further, Jiang et al. (2020) mentions that their results outperform three other solutions called *M01* (König et al., 1993; Rudeva and Gulev, 2007; Geng and Sugi, 2001), *M02* (Murray and Simmonds, 1991; Simmonds and Keay, 2000; Lim and Simmonds, 2007) and *M04* (Flaounas et al., 2014) which were published earlier. Other published algorithms do not provide their accuracy against manually created data sets. They usually demonstrate the accuracy of individual cases or measure the overall coherence of selected data.

### 3.2.2 Visual comparison

Two random sample images are shown in Fig. 6 and Fig. 7 so that we can compare our solution with data from *Met Office*. Red dots indicate high-pressure systems, whereas blue dots are used for low-pressure systems. The left part of each figure shows the results of the proposed solution in the equirectangular projection (the projection was adopted directly from the source of the dataset). The right part of each figure contains the *Met Office* data. The projection for this image is different but, the borders
of land and the range of latitude and longitude area are the same for both images.

In Fig. 6, we can notice a high-pressure system (#1) over Austria that is not present in the *Met Office* data.

In Fig. 7, our algorithm detected fewer systems: the missing systems are observed in the Atlantic ocean (#1), near Scotland (#2), and over Northern Africa (#3). Systems #1 and #2 are near the end of their lifetime. Thus, they are not recognized completely by the proposed algorithm. The difference in System #3 can be caused by the difference in the input dataset.

### 3.3 Performance

The performance of the algorithm is also important. If the algorithm runs fast, it can be used to reanalyze a large amount of data in a short time. In addition, it also makes it possible to modify the input parameters.



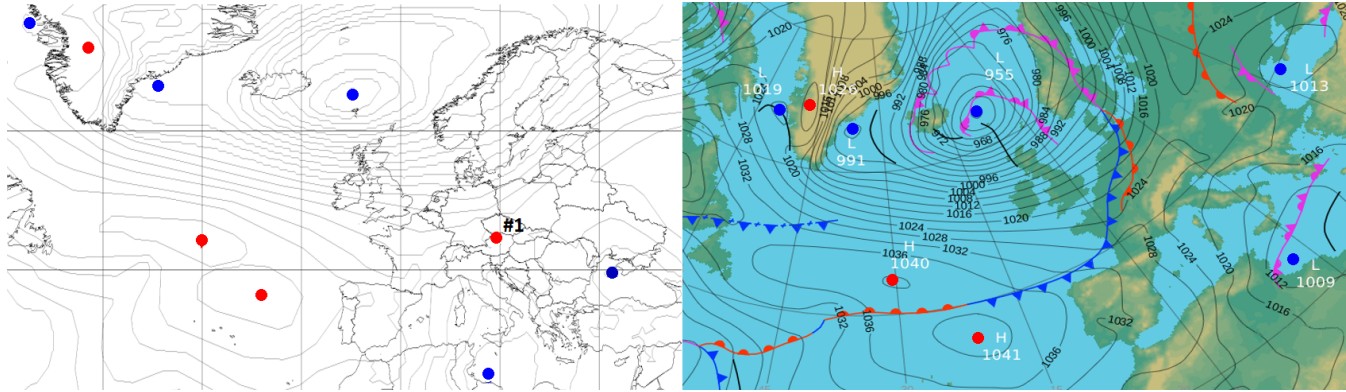

**Figure 6.** Comparison of detected pressure systems from the proposed solution (left) and *Met Office* (right). The red dots indicate high-pressure systems, and the blue dots represent low-pressure systems. The numbered systems differ in both figures. Data from 2014-12-10 12:00 UTC.

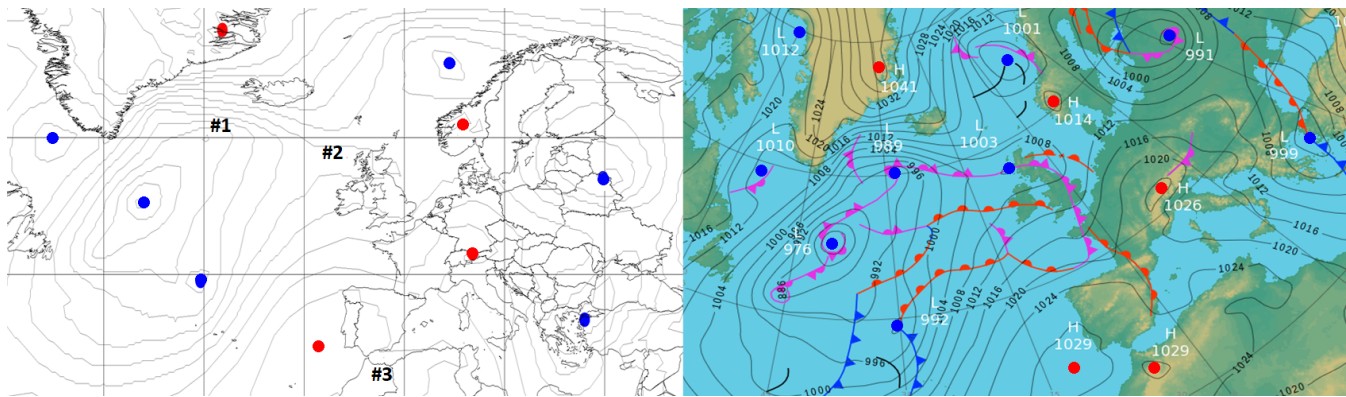

**Figure 7.** Comparison of detected pressure systems from the proposed solution (left) and *Met Office* (right). The red dots indicate high-pressure systems, and the blue dots represent low-pressure systems. The numbers denote systems missing in our solution. Data from 2021-01-27 00:00 UTC.

We have performed several tests with different resolutions and input coverage (global and local data). We have used an older CPU with 4.0GHz (Core i7-4790K) and 32GB RAM.

The averaged run-times with the standard deviation are available in Table 7. We can see that the performance of the algorithm is affected mainly by the number of detected pressure systems. The regional models contain fewer systems. In addition, their performance is better even though the resolution is similar to that of global models. The resolution also affects the performance. With the higher resolution, the run time is longer even though there is a similar number of detected systems.





**Table 7.** Measured average run-times in [ms] with a standard deviation $\sigma$ in [ms]. The column Systems denotes the average number of detected systems.

| Resolution | Coverage | Systems | Time [ms] | $\sigma$ [ms] |
|---|---|---|---|---|
| $1440 \times 721$ | Global | 84 | 1483 | 148 |
| $2880 \times 1441$ | Global | 95 | 1960 | 163 |
| $1094 \times 657$ | Europe | 13 | 249 | 9 |
| $1799 \times 1059$ | USA | 21 | 503 | 18 |

## 4 Conclusions

An algorithm for automatic detection of low- and high-pressure systems has been presented in the paper.

     The proposed solution uses only the MSLP layer for detection. This approach is similar to many existing solutions (Neu et al., 2013). Sometimes vorticity is used (e.g., Simmonds et al., 2008), which could improve identifying cyclones at a very early stage of development. However, vorticity is not suitable for determining high-pressure systems centers. Therefore, it cannot be deployed in our algorithm. Moreover, the presented solution does not need direct terrain filtering used by some other

algorithms. Our algorithm automatically discards many small systems that are usually scattered over higher mountains (e.g., the Himalayas, the Rocky Mountains, and the Andes).

     Our solution provides advantages in several areas. In general, it is easy to implement because it is based only on standard image processing algorithms. In addition, the performance of the proposed algorithm is sufficient for general use. Last but not least, not only can it detect low-pressure systems, as is usually standard by other algorithms, but also it can localize the centers

of high-pressure systems.

     However, it is challenging to validate the quality of results. The correct position of the pressure system cannot be determined exactly (see Neu et al., 2013). There is no universal definition that would precisely determine the center of a low- or high-pressure system. If systems are created manually, each author may prefer a slightly different position and properties of the system. This results in smaller or larger differences, which may partly be influenced by different data interpolations of the

analyzed field. Automated solutions, based on algorithms, share similar problems. Each solution is adjusted to detect systems that the authors of the algorithm consider important.

     Some studies (e.g., Neu et al., 2013) compare several algorithms. However, they conclude that there are no "correct" and "incorrect" results. All algorithms can be considered "correct", depending on the requirements that the system must meet. Based on these findings, our solution was not directly compared with the existing algorithms. Instead, the selection of 385 manually

created image-analysis by *Met Office*, which are widely excepted by the meteorologist, was compared. The comparison shows 83% through 92% agreements between the two approaches. This is quite similar successfulness to other methods (e.g., Dacre and Gray, 2009; Hanley and Caballero, 2012; Jiang et al., 2020; Neu et al., 2013). The most significant differences can be

found in the detection of cyclones at the beginning or end of the lifespan stage. *Met Office* meteorologists detect cyclones in these stages more often than does the presented solution.

The proposed solution requires to set up two parameters while the others can be auto-calculated. However, the need for input parameters can be found in other solutions as well (Raible et al., 2008). Based on our experiments, the algorithm is only a little sensitive to the values of input parameters. Most of the time, the number of detected systems is in a stable interval regardless of the values of the parameters. The problems with the detection of too many systems arise if the values are set too small.

    We provide a simple demo application of the proposed solution. Its source code is available at https://doi.org/10.5281/
zenodo.5163204 (Prantl, 2021).

    In the perspective of our further research, we will use the presented solution for a larger reanalysis of historical data in order to track changes of cyclones and anticyclones over the last decades. This will help us to analyze how the climate change affects the trajectories and frequencies of pressure systems.

*Code and data availability.* The code of the algorithms can be download from https://doi.org/10.5281/zenodo.5163204 (Prantl, 2021). The
data that support the findings of this study are ERA5 hourly data at single levels from 1979 to the present. They are openly available from Copernicus at http://doi.org10.24381/cds.adbb2d47 (Hersbach et al., 2021). Data for comparisons are obtained from *Met Office* and archived by *WetterZentrale*. They are publicly available at https://www.wetterzentrale.de/de/reanalysis.php?map=1&model=bra&var=45 (Müller and Floors, 2021)

*Author contributions.* MP designed and programmed the algorithm. MŽ provided expertise in meteorology and researched state-of-the-art
methods. DP carried out experiments and measurements. The paper has been written by MP, MŽ, and DP, and all authors have contributed to reviewing the text.

*Competing interests.* The authors declare that they have no conflict of interest.

*Acknowledgements.* This work was supported by the Department of Atmospheric Physics, Faculty of Mathematics and Physics, Charles University, and partially supported by the Czech Ministry of Education, Youth and Sports, project PUNTIS (LO1506)





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
