# Peer review of "CycloneDetector (v1.0) - Algorithm for detecting cyclone and anticyclone centers from mean sea level pressure layer"

_Geoscientific Model Development, 2021_

## Referee Comment (RC2)

Review for gmd-2021-266

Cyclone Detector (v1.0) - Algorithm for detecting cyclone and anticyclone centers from mean sea level pressure layer

The authors present in this manuscript a cyclone (anticyclone) tracking algorithm based on SLP analysis. Although the variable has been widely used in other studies of cyclones, the method could be interesting. According to the manuscript, the method can be summarized as follows: search the closed isobars and in order to remove unwanted areas, a filtering based on the area of the contour area is used with some considerations about the projection of the areas (AABB). Perhaps we could say that the method has somehow, similarities with a clustering approach using isobars as centroids (Kmeans). Such a method might not be entirely accurate in the case of multiple cyclones in small areas, particularly when they may have different lifetimes and seems to underestimate the complexity of these events and their dynamics.

I find, according to what is expressed in the manuscript, that the algorithm needs decisions in several steps that are somewhat arbitrary and / or depend on the user. For example, the area "T" enclosed is user-selectable, the threshold value can be adjusted by the user, the isobar spacing etc. I don't think that an algorithm for cyclone detection should strongly depend on users. This could lead to very dissimilar results depending on the options chosen, and therefore not valid for comparisons. Precisely, the idea of developing powerful algorithms to track and study these events should contribute to producing statistics and analyses that are increasingly detailed and comparable with others already made (be these based on SLP, vorticity, etc).

It is also important, as the referee # 1 mentioned, that the dependence (no dependence) of the projections used and the results (if no evidence can be show to support it), should be at least tested with a wide number of events. I believe that more powerful and well-founded arguments are needed to support this idea.

I must also agree with the previous referee that the analysis of the results is simple and at least insufficient. The idea of validating the algorithm should cover broader areas of study (there are all over the world), and not just be based on number of events.

There are several methods used to track cyclones and some recently published. In these methods, the algorithm is automated and also provides substantial information regarding not only the position of these events but also lifetimes and trajectories.

I believe that the method proposed here, even though the idea could be interesting, should be improved and completed in order to contribute to the methods already in use, that in addition to detecting low and high pressure centers, can provide trajectories, and lifetimes (very important characteristics for a cyclone analysis)